# YASS: Yet Another Spike Sorter

JinHyung Lee[1], David Carlson[2], Hooshmand Shokri[1], Weichi Yao[1], Georges Goetz[3], Espen Hagen[4], Eleanor Batty[1], EJ Chichilnisky[3], Gaute Einevoll[5], and Liam Paninski[1]

[1]Columbia University, [2]Duke University, [3]Stanford University, [4]University of Oslo, [5]Norwegian University of Life Sciences

## Abstract

Spike sorting is a critical first step in extracting neural signals from large-scale electrophysiological data. This manuscript describes an efficient, reliable pipeline for spike sorting on dense multi-electrode arrays (MEAs), where neural signals appear across many electrodes and spike sorting currently represents a major computational bottleneck. We present several new techniques that make dense MEA spike sorting more robust and scalable. Our pipeline is based on an efficient multi-stage "triage-then-cluster-then-pursuit" approach that initially extracts only clean, high-quality waveforms from the electrophysiological time series by temporarily skipping noisy or "collided" events (representing two neurons firing synchronously). This is accomplished by developing a neural network detection method followed by efficient outlier triaging. The clean waveforms are then used to infer the set of neural spike waveform templates through nonparametric Bayesian clustering. Our clustering approach adapts a "coreset" approach for data reduction and uses efficient inference methods in a Dirichlet process mixture model framework to dramatically improve the scalability and reliability of the entire pipeline. The "triaged" waveforms are then finally recovered with matching-pursuit deconvolution techniques. The proposed methods improve on the state-of-the-art in terms of accuracy and stability on both real and biophysically-realistic simulated MEA data. Furthermore, the proposed pipeline is efficient, learning templates and clustering faster than real-time for a $\simeq 500$-electrode dataset, largely on a single CPU core.

## 1 Introduction

The analysis of large-scale multineuronal spike train data is crucial for current and future neuroscience research. These analyses are predicated on the existence of reliable and reproducible methods that feasibly scale to the increasing rate of data acquisition. A standard approach for collecting these data is to use dense multi-electrode array (MEA) recordings followed by "spike sorting" algorithms to turn the obtained raw electrical signals into spike trains.

A crucial consideration going forward is the ability to scale to massive datasets: MEAs currently scale up to the order of $10^4$ electrodes, but efforts are underway to increase this number to $10^6$ electrodes[1]. At this scale any manual processing of the obtained data is infeasible. Therefore, automatic spike sorting for dense MEAs has enjoyed significant recent attention [15, 9, 51, 24, 36, 20, 33, 12]. Despite these efforts, spike sorting remains the major computational bottleneck in the scientific pipeline when using dense MEAs, due both to the high computational cost of the algorithms and the human time spent on manual postprocessing.

To accelerate progress on this critical scientific problem, our proposed methodology is guided by several main principles. First, *robustness* is critical, since hand-tuning and post-processing is not

**Algorithm 1** Pseudocode for the complete proposed pipeline.

---

Input: time-series of electrophysiological data $\mathbf{V} \in \mathbb{R}^{T \times C}$, locations $\in \mathbb{R}^3$
$[\text{waveforms}, \text{timestamps}] \leftarrow \texttt{Detection}(\mathbf{V})$ % (Section 2.2)
% "Triage" noisy waveforms and collisions (Section 2.4):
$[\text{cleanWaveforms}, \text{cleanTimestamps}] \leftarrow \texttt{Triage}(\text{waveforms}, \text{timestamps})$
% Build a set of representative waveforms and summary statistics (Section 2.5)
$[\text{representativeWaveforms}, \text{sufficientStatistics}] \leftarrow \texttt{coresetConstruction}(\text{cleanWaveforms})$
% DP-GMM clustering via divide-and-conquer (Sections 2.6 and 2.7)
$[\{\text{representativeWaveforms}_i, \text{sufficientStatistics}_i\}_{i=1,...}]$
    $\leftarrow \texttt{splitIntoSpatialGroups}(\text{representativeWaveforms}, \text{sufficientStatistics}, \text{locations})$
**for** i=1,... **do** % Run efficient inference for the DP-GMM
    $[\text{clusterAssignments}_i] \leftarrow \texttt{SplitMergeDPMM}(\text{representativeWaveforms}_i, \text{sufficientStatistics}_i)$
**end for**
% Merge spatial neighborhoods and similar templates
$[\text{allClusterAssignments}, \text{templates}] \leftarrow$
    $\texttt{mergeTemplates}(\{\text{clusterAssignments}_i\}_{i=1,...}, \{\text{representativeWaveforms}_i\}_{i=1,...}, \text{locations})$
% Pursuit stage to recover collision and noisy waveforms
$[\text{finalTimestamps}, \text{finalClusterAssignments}] \leftarrow \texttt{deconvolution}(\text{templates})$
**return** $[\text{finalTimestamps}, \text{finalClusterAssignments}]$

---

feasible at scale. Second, *scalability* is key. To feasibly process the oncoming data deluge, we use efficient data summarizations wherever possible and focus computational power on the "hard cases," using cheap fast methods to handle easy cases. Next, the pipeline should be *modular*. Each stage in the pipeline has many potential feasible solutions, and the pipeline is improved by rapidly iterating and updating each stage as methodology develops further. Finally, *prior information* is leveraged as much as possible; we share information across neurons, electrodes, and experiments in order to extract information from the MEA datastream as efficiently as possible.

We will first outline the methodology that forms the core of our pipeline in Section 2.1, and then we demonstrate the improvements in performance on simulated data and a 512-electrode recording in Section 3. Further supporting results appear in the appendix.

## 2 Methods

### 2.1 Overview

The inputs to the pipeline are the band-pass filtered voltage recordings from all $C$ electrodes and their spatial layout, and the end result of the pipeline is the set of $K$ (where $K$ is determined by the algorithm) binary neural spike trains, where a "1" in the time series reflects a neural action potential from the $k$th neuron at the corresponding time point. The voltage signals are spatially whitened prior to processing and are modeled as the superposition of action potentials and background Gaussian noise [12]. Spatial whitening is performed by removing potential spikes using amplitude thresholding and estimating the whitening filter under a Gaussianity assumption. Succinctly, the pipeline is a multistage procedure as follows: ($i$) detecting waveforms and extracting features, ($ii$) screening outliers and collided waveforms, ($iii$) clustering, and ($iv$) inferring missed and collided spikes. Pseudocode for the flow of the pipeline can be found in Algorithm 1. A brief overview is below, followed by additional details.

Our overall strategy can be considered a hybrid of a matching pursuit approach (similar to that employed by [36]) and a classical clustering approach, generalized and adapted to the large dense MEA setting. Our guiding philosophy is that it is essential to properly handle "collisions" between simultaneous spikes [37, 12], since collisions distort the extracted feature space and hinder clustering. A typical approach to this issue utilizes matching pursuit methods (or other sparse deconvolution strategies), but these methods are relatively computationally expensive compared to clustering primitives. This led us to a "triage-then-cluster-then-pursuit" approach: we "triage" collided or overly noisy waveforms, putting them aside during the feature extraction and clustering stages, and later recover these spikes during a final "pursuit" or deconvolution stage. The triaging begins during the detection stage in Section 2.2, where we develop a neural network based detection method that

significantly improves sensitivity and selectivity. For example, on a simulated 30 electrode dataset with low SNR, the new approach reduces false positives and collisions by 90% for the same rate of true positives. Furthermore, the neural network is significantly better at the *alignment* of signals, which improves the feature space and signal-to-noise power. The detected waveforms then are projected to a feature space and restricted to a local spatial subset of electrodes as in [24] in Section 2.3. Next, in Section 2.4 an outlier detection method further "triages" the detected waveforms and reduces false positives and collisions by an additional 70% while removing only a small percentage of real detections. All of these steps are achievable in nearly linear time. Simulations demonstrate that this large reduction in false positives and collisions dramatically improves accuracy and stability.

Following the above steps, the remaining waveforms are partitioned into distinct neurons via clustering. Our clustering framework is based on the Dirichlet Process Gaussian Mixture Model (DP-GMM) approach [48, 9], and we modify existing inference techniques to improve scalability and performance. Succinctly, each neuron is represented by a distinct Gaussian distribution in the feature space. Directly calculating the clustering on all of the channels and all of the waveforms is computationally infeasible. Instead, the inference first utilizes the spatial locality via *masking* [24] from Section 2.3. Second, the inference procedure operates on a *coreset* of representative points [13] and the resulting approximate sufficient statistics are used in lieu of the full dataset (Section 2.5). Remarkably, we can reduce a dataset with $100k$ points to a coreset of $\simeq 10k$ points with trivial accuracy loss. Next, split and merge methods are adapted to efficiently explore the clustering space [21, 24] in Section 2.6. Using these modern scalable inference techniques is crucial for *robustness* because they empirically find much more sensible and accurate optima and permit Bayesian characterization of posterior uncertainty.

For very large arrays, instead of operating on all channels simultaneously, each distinct spatial neighborhood is processed by a separate clustering algorithm that may be run in parallel. This parallelization is crucial for processing very large arrays because it allows greater utilization of computer resources (or multiple machines). It also helps improve the efficacy of the split-merge inference by limiting the search space. This divide-and-conquer approach and the post-processing to stitch the results together is discussed in Section 2.7. The computational time required for the clustering algorithm scales nearly linearly with the number of electrodes $C$ and the experiment time.

After the clustering stage is completed, the means of clusters are used as templates and collided and missed spikes are inferred using the deconvolution (or "pursuit" [37]) algorithm from Kilosort [36], which recovers the final set of binary spike trains. We limit this computationally expensive approach only to sections of the data that are not well handled by the rest of the pipeline, and use the faster clustering approach to fill in the well-explained (i.e. easy) sections.

We note finally that when memory is limited compared to the size of the dataset, the preprocessing, spike detection, and final deconvolution steps are performed on temporal minibatches of data; the other stages operate on significantly reduced data representations, so memory management issues typically do not arise here. See Section B.4 for further details on memory management.

## 2.2 Detection

The detection stage extracts temporal and spatial windows around action potentials from the noisy raw electrophysiological signal $\mathbf{V}$ to use as inputs in the following clustering stage. The number of *clean* waveform detections (true positives) should be maximized for a given level of detected collision and noise events (false positives). Because collisions corrupt feature spaces [37, 12] and will simply be recovered during pursuit stage, they are not included as true positives at this stage. In contrast to the plethora of prior work on hand-designed detection rules (detailed in Section C.1), we use a data-driven approach with neural networks to dramatically improve both detection efficacy and alignment quality. The neural network is trained to return only clean waveforms on real data, not collisions, so it *de facto* performs a preliminary triage prior to the main triage stage in Section 2.4.

The crux of the data-driven approach is the availability of prior training data. We are targeting the typical case that an experimental lab performs repeated experiments using the same recording setup from day to day. In this setting hand-curated or otherwise validated prior sorts are saved, resulting in abundant training data for a given experimental preparation. In the supplemental material, we discuss the construction of a training set (including data augmentation approaches) in Section C.2, the architecture and training of the network in Section C.3, the detection using the network in Section C.4, empirical performance in Section C.5, and scalability in Section C.5. This strategy is effective when

this training data exists; however, many research groups are currently using single electrodes and do not have dense MEA training data. Thus it is worth emphasizing that here we train the detector only on a single electrode. We have also experimented with training and evaluating on multiple electrodes with good success; however, these results are more specialized and are not shown here.

A key result is that our neural network dramatically improves both the temporal and spatial alignment of detected waveforms. This improved alignment improves the fidelity of the feature space and the signal-to-noise power, and the result of the improved feature space can clearly be seen by comparing the detected waveform features from one standard detection approach (`SpikeDetekt` [24]) in Figure 1 (left) to the detected waveform features from our neural network in Figure 1 (middle). Note that the output of the neural net detection is remarkably more Gaussian compared to SpikeDetekt.

## 2.3 Feature Extraction and Mask Creation

Following detection we have a collection of $N$ events defined as $\mathbf{X}_n \in \mathbb{R}^{R \times C}$ for $n = 1, \ldots, N$, each with an associated detection time $t_n$. Recall that $C$ is the total number of electrodes, and $R$ is the number of time samples, in our case chosen to correspond to 1.5ms. Next features are extracted by using uncentered Principal Components Analysis (PCA) on each channel separately with $P$ features per channel. Each waveform $\mathbf{X}_n$ is transformed to the feature space $\mathbf{Y}_n$. To handle duplicate spikes, $\mathbf{Y}_n$ is kept only if $c_n = \arg\max\{||\mathbf{y}_{nc}||_{c \in \mathcal{N}_{c_n}}\}$, where $\mathcal{N}_{c_n}$ contains all electrodes in the local neighborhood of electrode $c_n$. To address the increasing dimensionality, spikes are localized by using the sparse masking vector $\{\mathbf{m}_n\} \in [0, 1]^C$ method of [24], where the mask should be set to 1 only where the signal exists. The sparse vector reduces the dimensionality and facilitates sparse updates to improve computational efficiency. We give additional mathematical details in Supplemental Section D. We have also experimented with an autoencoder framework to standardize the feature extraction across channels and facilitate online inference. This approach performed similarly to PCA and is not shown here, but will be addressed in depth in future work.

## 2.4 Collision Screening and Outlier Triaging

Many collisions and outliers remain even after our improved detection algorithm. Because these events destabilize the clustering algorithms, the pipeline benefits from a "triage" stage to further screen collisions and noise events. Note that triaging out a small fraction of true positives is a minor concern at this stage because they will be recovered in the final deconvolution step.

We use a two-fold approach to perform this triaging. First, obvious collisions with nearly overlapping spike times and spatial locations are removed. Second, k-Nearest Neighbors (k-NN) is used to detect outliers in the masked feature space based on [27]. To develop a computationally efficient and effective approach, waveforms are grouped based on their primary (highest-energy) channel, and then k-NN is run for each channel. Empirically, these approximations do not suffer in efficacy compared to using the full spatial area. When the dimensionality of $P$, the number of features per channel, is low, a kd-tree can find neighbors in $\mathcal{O}(N \log N)$ average time. We demonstrate that this method is effective for triaging false positives and collisions in Figure 1 (middle).

## 2.5 Coreset Construction

"Big data" improves density estimates for clustering, but the cost per iteration naively scales with the amount of data. However, often data has some redundant features, and we can take advantage of these redundancies to create more efficient summarizations of the data. Then running the clustering algorithm on the summarized data should scale only with the number of summary points. By choosing representative points (or a "coreset") carefully we can potentially describe huge datasets accurately with a relatively small number of points [19, 13, 2].

There is a sizable literature on the construction of coresets for clustering problems; however, the number of required representative points to satisfy the theoretical guarantees is infeasible in this problem domain. Instead, we propose a simple approach to build coresets that empirically outperforms existing approaches in our experiments by forcing adequate coverage of the complete dataset. We demonstrate in Supplemental Figure S6 that this approach can cover clusters completely missed by existing approaches, and show the chosen representative points on data in Figure 1 (right). This algorithm is based on recursively performing k-means; we provide pseudocode and additional details

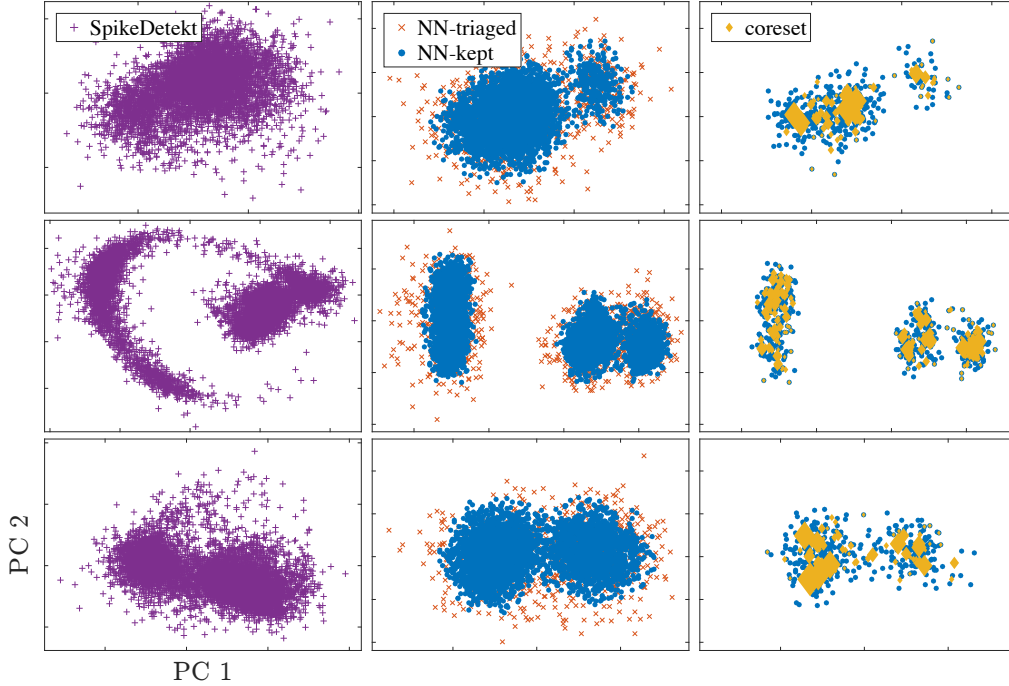

Figure 1: **Illustration of Neural Network Detection, Triage, and Coreset from a primate retinal ganglion cell recording.** The first column shows spike waveforms from `SpikeDetekt` in their PCA space. Due to poor alignment, clusters have a non-Gaussian shape with many outliers. The second column shows spike waveforms from our proposed neural network detection in the PCA space. After triaging outliers, the clusters have cleaner Gaussian shapes in the recomputed feature space. The last column illustrates the coreset. The size of each coreset diamond represents its weight. For visibility, only 10% of data are plotted.

in in Supplemental Section E. The worst case time complexity is nearly linear with respect to the number of representative points, the number of detected spikes, and the number of channels. The algorithm ends by returning $G$ representative points, their sufficient statistics, and masks.

## 2.6 Efficient Inference for the Dirichlet Process Gaussian Mixture Model

For the clustering step we use a Dirichlet Process Gaussian Mixture Model (DP-GMM) formulation, which has been previously used in spike sorting [48, 9], to adaptively choose the number of mixture components (visible neurons). In contrast to these prior approaches, here we adopt a Variational Bayesian split-merge approach to explore the clustering space [21] and to find a more robust and higher-likelihood optimum. We address the high computational cost of this approach with several key innovations. First, following [24], we fit a mixture model on the virtual masked data to exploit the localized nature of the data. Second, following [9, 24], the covariance structure is approximated as a block-diagonal to reduce the parameter space and computation. Finally, we adapted the methodology to work with the representative points (coreset) rather than the raw data, resulting in a highly scalable algorithm. A more complete description of this stage can be found in Supplemental Section F, with pseudocode in Supplemental Algorithm S2.

In terms of computational costs, the dominant cost per iteration in the DPMM algorithm is the calculation of data to cluster assignments, which in our framework will scale at $\mathcal{O}(G\bar{m}P^2K)$, where $\bar{m}$ is the average number of channels maintained in the mask for each of the representative points, $G$ is the number of representative points, and $P$ is the number of features per channel. This is in stark contrast to a scaling of $\mathcal{O}(NC^2P^2K + P^3)$ without our above modifications. Both $K$ and $G$ are expected to scale linearly with the number of electrodes and *sublinearly* with the length of the recording. Without further modification, the time complexity in the above clustering algorithm would depend on the square of the number of electrodes for each iteration; fortunately, this can be reduced to a linear dependency based on a divide-and-conquer approach discussed below in Section 2.7.

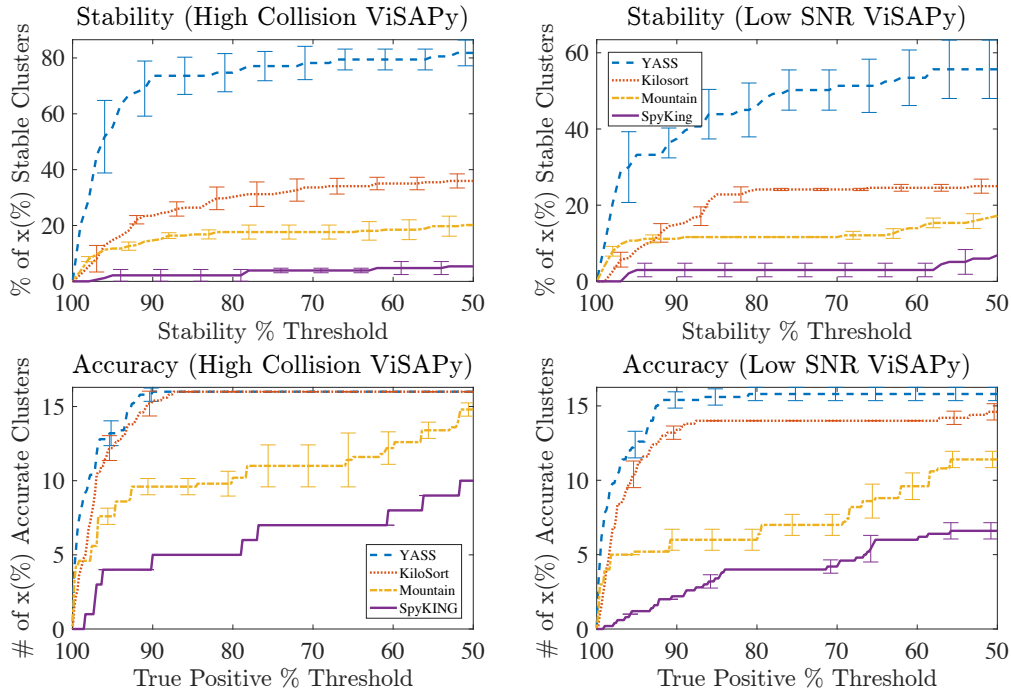

Figure 2: **Simulation results on 30-channel `ViSAPy` datasets.** Left panels show the result on `ViSAPy` with high collision rate and Right panels show the result on `ViSAPy` with low SNR setting. (Top) stability metric (following [5]) and percentage of total discovered clusters above a certain stability measure. The noticeable gap between stability of `YASS` and the other methods results from a combination of high number of stable clusters and lower number of total clusters. (Bottom) These results show the number of clusters (out of a ground truth of 16 units) above a varying quality threshold for each pipeline. For each level of accuracy, the number of clusters that pass that threshold is calculated to demonstrate the relative quality of the competing algorithms on this dataset. Empirically, our pipeline (`YASS`) outperforms other methods.

## 2.7 Divide and Conquer and Template Merging

Neural action potentials have a finite spatial extent [6]. Therefore, the spikes can be divided into distinct groups based on the geometry of the MEA and the local position of each neuron, and each group is then processed independently. Thus, each group can be processed in parallel, allowing for high data throughput. This is crucial for exploiting parallel computer resources and limited memory structures. Second, the split-and-merge approach in a DP-GMM is greatly hindered when the numbers of clusters is very high [21]. The proposed divide and conquer approach addresses this problem by greatly reducing the number of clusters within each subproblem, allowing the split and merge algorithm to be targeted and effective.

To divide the data based on the spatial location of each spike, the primary channel $c_n$ is determined for every point in the coreset based on the channel with maximum energy, and clustering is applied on each channel. Because neurons may now end up on multiple channels, it is necessary to merge templates from nearby channels as a post-clustering step. When the clustering is completed, the mean of each cluster is taken as a template. Because waveforms are limited to their primary channel, some neurons may have "overclustered" and have a distinct mixture component on distinct channels. Also, overclustering can occur from model mismatch (non-Gaussianity). Therefore, it is necessary to merge waveforms. Template merging is performed based on two criteria, the angle and the amplitude of templates, using the best alignment on all temporal shifts between two templates. The pseudocode to perform this merging is shown in Supplemental Algorithm S3. Additional details can be found in Supplemental Section G.

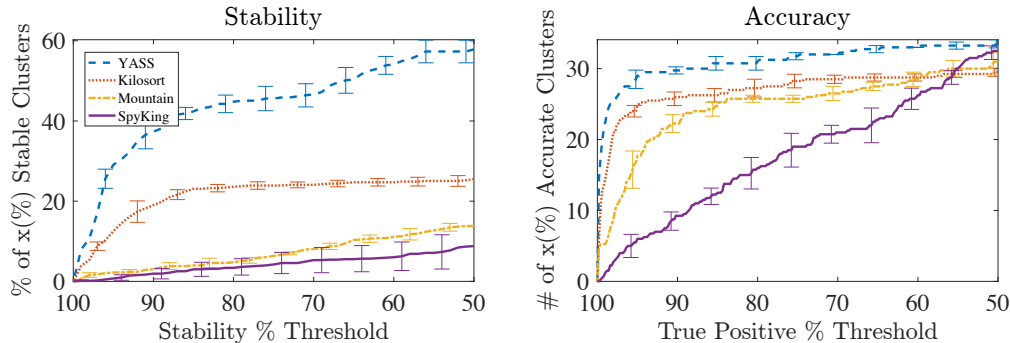

Figure 3: **Performance comparison of spike sorting pipelines on primate retina data.** (Left) The same type of plot as in the top panels of Figure 2. (Right) The same type of plot as in the bottom panels of Figure 2 compared to the "gold standard" sort. YASS demonstrates both improved stability and also per-cluster accuracy.

## 2.8 Recovering Triaged Waveforms and Collisions

After the previous steps, we apply matching pursuit [36] to recover triaged waveforms and collisions. We detail the available choices for this stage in Supplemental Section I.

## 3 Performance Comparison

We evaluate performance to compare several algorithms (detailed in Section 3.1) to our proposed methodology on both synthetic (Section 3.2) and real (Section 3.3) dense MEA recordings. For each synthetic dataset we evaluate the ability to capture ground truth in addition to the per-cluster stability metrics. For the ground truth, inferred clusters are matched with ground truth clusters via the Hungarian algorithm, and then the per-cluster accuracy is calculated as the number of assignments shared between the inferred cluster and the ground truth cluster over the total number of waveforms in the inferred cluster. For the per-cluster stability metric, we use the method from Section 3.3 of [5] with the rate scaling parameter of the Poisson processes set to 0.25. This method evaluates how robust individual clusters are to perturbations of the dataset. In addition, we provide runtime information to empirically evaluate the computational scaling of each approach. The CPU runtime was calculated on a single core of a six-core i7 machine with 32GB of RAM. GPU runtime is given from a Nvidia Titan X within the same machine.

## 3.1 Competing Algorithms

We compare our proposed pipeline to three recently proposed approaches for dense MEA spike sorting: KiloSort [36], Spyking Circus [51], and MountainSort [31]. Kilosort, Spyking Cricus, and MountainSort were downloaded on January 30, 2017, May 26th, 2017, and June 7th, 2017, respectively. We dub our algorithm Yet Another Spike Sorter (YASS). We discuss additional details on the relationships between these approaches and our pipeline in Supplemental Section I. All results are shown with no manual post-processing.

## 3.2 Synthetic Datasets

First, we used the biophysics-based spike activity generator ViSAPy [18] to generate multiple 30-channel datasets with different noise levels and collision rates. The detection network was trained on the ground truth from a low signal-to-noise level recording. Then, the trained neural network is applied to all signal-to-noise levels. The neural network dramatically outperforms existing detection methodologies on these datasets. For a given level of true positives, the number of false positives can be reduced by an order of magnitude. The properties of the learned network are shown in Supplemental Figures S4 and S5.

Performance is evaluated on the known ground truth. For each level of accuracy, the number of clusters that pass that threshold is calculated to demonstrate the relative quality of the competing

| Detection (GPU) | Data Ext. | Triage | Coreset | Clustering | Template Ext. | Total |
|---|---|---|---|---|---|---|
| 1m7s | 42s | 11s | 34s | 3m12s | 54s | 6m40s |

Table 1: **Running times of the main processes on 512-channel primate retinal recording of 30 minutes duration.** Results shown using a single CPU core, except for the detection step (2.2), which was run on GPU. We found that full accuracy was achieved after processing just one-fifth of this dataset, leading to significant speed gains. Data Extraction refers to waveform extraction and Performing PCA (2.3). Triage, Coreset, and Clustering refer to 2.4, 2.5, and 2.6, respectively. Template Extraction describes revisiting the recording to estimate templates and merging them (2.7). Each step scales approximately linearly (Section B.3).

algorithms on this dataset. Empirically, our pipeline (YASS) outperforms other methods. This is especially true in low SNR settings, as shown in Figure 2. The per-cluster stability metric is also shown in Figure 2. The stability result demonstrates that YASS has significantly fewer low-quality clusters than competing methods.

### 3.3   Real Datasets

To examine real data, we focused on 30 minutes of extracellular recordings of the peripheral primate retina, obtained ex-vivo using a high-density 512-channel recording array [30]. The half-hour recording was taken while the retina was stimulated with spatiotemporal white noise. A "gold standard" sort was constructed for this dataset by extensive hand validation of automated techniques, as detailed in Supplemental Section H. Nonstationarity effects (time-evolution of waveform shapes) were found to be minimal in this recording (data not shown).

We evaluate the performance of YASS and competing algorithms using 4 distinct sets of 49 spatially contiguous electrodes. Note that the gold standard sort here uses the information from the full 512-electrode array, while we examine the more difficult problem of sorting the 49-electrode data; we have less information about the cells near the edges of this 49-electrode subset, allowing us to quantify the performance of the algorithms over a range of effective SNR levels. By comparing the inferred results to the gold standard, the cluster-specific true positives are determined in addition to the stability metric. The results are shown in Figure 3 for one of the four sets of electrodes, and the remaining three sets are shown in Supplemental Section B.1. As in the simulated data, compared to KiloSort, which had the second-best overall performance on this dataset, YASS has dramatically fewer low-stability clusters.

Finally, we evaluate the time required for each step in the YASS pipeline (Table 1). Importantly, we found that YASS is highly robust to data limitations: as shown in Supplemental Figure S3 and Section B.3, using only a fraction of the 30 minute dataset has only a minor impact on performance. We exploit this to speed up the pipeline. Remarkably, running primarily on a single CPU core (only the detect step utilizes a GPU here), YASS achieves a several-fold speedup in template and cluster estimation compared to the next fastest competitor[2], Kilosort, which was run in full GPU mode and spent about 30 minutes on this dataset. We plan to further parallelize and GPU-ize the remaining steps in our pipeline next, and expect to achieve significant further speedups.

## 4   Conclusion

YASS has demonstrated state-of-the-art performance in accuracy, stability, and computational efficiency; we believe the tools presented here will have a major practical and scientific impact in large-scale neuroscience. In our future work, we plan to continue iteratively updating our modular pipeline to better handle template drift, refractory violations, and dense collisions.

Lastly, YASS is available online at `https://github.com/paninski-lab/yass`

**Acknowledgements**

This work was partially supported by NSF grants IIS-1546296 and IIS-1430239, and DARPA Contract No. N66001-17-C-4002.

## Footnotes

[1]DARPA Neural Engineering System Design program BAA-16-09

[2]Spyking Circus took over a day to process this dataset. Assuming linear scaling based on smaller-scale experiments, Mountainsort is expected to take approximately 10 hours.

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
