[Supplementary Material · yass-spike-sorter_supplement.pdf]

| Notation | Explanation | Default value (if exists) |
|---|---|---|
| **Data Constants** | | |
| $T$ | Recording length | |
| $C$ | Total number of channels | |
| $N$ | Number of detected spikes | |
| $t_n$ | Temporal location of spike $n$ | |
| $c_n$ | Index of the main channel of spike $n$ | |
| $C_{eff}$ | Number of neighboring channels (including the main channel) | |
| $G$ | Number of representative points after the Coreset algorithm | |
| | | |
| **Data structures** | | |
| $\mathbf{V} \in \mathbb{R}^{T \times C}$ | Recording | |
| $\mathbf{X_n} \in \mathbb{R}^{R \times C}$ | Voltage trace (waveform) of spike $n$ | |
| $\mathbf{m_n} \in [0,1]^C$ | Masking vector for spike $n$ | |
| $\mathbf{Y_n} \in \mathbb{R}^{P \times C}$ | $\mathbf{X_n}$ mapped to the feature (PCA) space | |
| $\mathbf{x}_{nc}, \mathbf{y}_{nc}$ | $c^{\text{th}}$ column of $\mathbf{X}_n, \mathbf{Y}_n$ | |
| $m_{nc}$ | $c^{\text{th}}$ entry of $\mathbf{m}_m$ | |
| $\tilde{\mathbf{X}}_\mathbf{n}, \tilde{\mathbf{x}}_{nc}, \tilde{\mathbf{Y}}_\mathbf{n}, \tilde{\mathbf{y}}_{nc}$ | Virtual data distribution of $\mathbf{X}_n, \mathbf{x}_{nc}, \mathbf{Y}_n, \mathbf{y}_{nc}$ | |
| $\mathbf{P} \in \mathbb{R}^{T \times C}$ | Probability output of NN | |
| $\mathbf{W_k} \in \mathbb{R}^{R \times C}$ | Template of cluster $k$ | |
| $\mathbf{m}_k^w \in \{0,1\}^C$ | Mask of cluster $k$ | |
| | | |
| **Parameters** | | |
| $R$ | Temporal window size of spike | 1.5 ms |
| $R'$ | Half of window size | |
| $P$ | Per channel dimension of data in PCA domain | 3 |
| $\tau$ | probability threshold for NN detection $\mathbf{P}$ | 0.5 |
| $\theta_w, \theta_s$ | weak and strong threshold for masking | $F_{\chi^2,P}^{-1}(0.5), F_{\chi^2,P}^{-1}(0.9)$ |
| $F_{\chi^2,df}^{-1}$ | Inverse CDF of Chi-squared Distribution with $df$ d.f. | |
| $K_{++}$ | Number of Clusters in Kmeans++ for Coreset | 10 |
| $D_{max}$ | Distance threshold for Coreset | $2F_{\chi^2,PC_{eff}}^{-1}(0.9)$ |
| $m_0, \lambda_0, \mathbf{W_0}, v_0$ | Prior parameters for the DP-GMM Normal-Whishart | $\mathbf{0}, 0.01, \frac{1}{v_0}\mathbf{I}_P, P+2$ |
| $\alpha_0$ | Prior parameter for beta dist. in stick-breaking in DP-GMM | 1 |
| $s_{max}$ | Maximum shift allowance for template merging | 0.5 ms |
| $\theta_1$ | threshold on cosine angle in template merging | 0.85 |
| $\theta_2$ | threshold on size in template merging | 0.6 |

Table S1: **Summary table of notation used within the manuscript.**

## A   Notation

The following notation is employed: scalars are lowercase italicized letters, e.g. $x$, constants such as max indices are represented by uppercase italicized letters, e.g. $N$, vectors are bolded lowercase letters, e.g. $\mathbf{x}$, and matrices are bolded uppercase letters, e.g. $\mathbf{X}$. Major notations used in the paper are summarized in Table S1.

## B   Additional results

In this section, we first provide the performance metrics on the other three sets of 49-channel recordings of the primate retina referenced from Section 3.3. Next, we describe how limiting the temporal duration of the data effects performance.

Figure S1: **Comparison of performance on the other three 49 channel datasets from primate retina.** Each row corresponds to one of the three additional datasets. Figures on the left depict stability metric and what percentage of total discovered clusters are above the chosen stability threshold on the x axis. Figures on the right depict the true positive accuracies with respect to partial ground truth and how many discovered clusters are above the chosen true positive accuracy.

## B.1 Results on Three Additional Real 49-Electrode Sets of Data

The summary performance metrics for the three additional recordings of the primate retina are reported in Figure S1. Note that `YASS` outperforms the other methods, especially in terms of stability of the clusters. This is due to `YASS` providing both more stable clusters and fewer total clusters in general.

## B.2 Effect of Each Module on YASS's Performance

The state-of-the-art performance of `YASS` is the result of advances in many stages of a complete pipeline. To understand the effect of each module, the performance of altering single stages of `YASS` is measured. Specifically, `YASS` is broken down by (i) replacing the Neural Network detection algorithm with `SpikeDetekt` [24], skipping (ii) outlier triaging, and skipping (iii) the coreset construction. The results are shown in Figure S2. As shown, altering these stages downgrades the overall performance of `YASS`.

Figure S2: **Accuracy comparison to incomplete** YASS This plot shows the same metrics as the right panel of Figure 3. However, it compares to versions of YASS with single stages altered. Here, YASS refers to the full pipeline, and the other approaches are as described in Section B.2. As illustrated, every part of YASS contributes to the overall performance.

## B.3 Accuracy with Respect to Data Length

The effect of using a partial recording to estimate waveform templates is investigated further. The summary is described in the left and right panels of Figure S3. To illustrate the effect, the full recording is randomly subsetted by the specified length. Waveform templates are estimated using information only from the subset. Accuracy loss becomes insignificant with more than 20% of the full length on a 30 minute dataset.

Furthermore, the scalability of YASS is demonstrated in the bottom panel of Figure S3. As illustrated, the main parts of the full pipeline, coreset and clustering, scale almost linearly.

## B.4 When Data Exceeds Memory

Large recordings exceed the memory capacity of typical workstations. This issue in handled in the preprocessing and spike detection by temporally partitioning the recording and processing each temporal subsection individually. Afterwards, the divide and conquer approach from Section 2.7 significantly reduces the memory requirement by reducing the number of waveforms and their spatial extent. If the memory limits are still exceeded due to extremely long recordings, data are randomly subsetted and postprocessed.

Another possible approach is that, following the result of Section B.3, data can be processed partially up to the point where it does not exceed memory. Once the templates are estimated based on the partial recording, deconvolution should handle spikes from the remainder of recording. We have not yet needed to implement this approach, however.

# C Additional details on the Detection Algorithm

## C.1 Relationships to Existing Detection Algorithms

The goal of a detection algorithm within a spike sorting pipeline is to extract (unsorted) action potentials from the raw electrophysiological signal to use as inputs for a downstream clustering algorithms. It is crucial for the subsequent steps of the pipeline that the detected action potentials cover all present neural shapes with few false positives, where false positives here are defined as either noise events, collisions (two or more waveforms simultaneously occurring in time or space), or poorly aligned spikes. Historically, most research labs have used a simple voltage threshold to determine whether a section of signal should be considered an action potential [29], but many other decision rules have been considered, such as the nonlinear energy operator [32] and wavelet thresholding [50].

Most proposed detection rules above operated on a single channel at a time (although Bayesian optimal detection has been used on multiple channels [8, 33]). A simple approach that works either

Figure S3: **Using only a portion of data to estimate templates.** (Top) YASS is tested on two sets of recording with 49 channels and 30 minutes length from the retinal cells. Only a portion of data is randomly extracted and templates are estimated. As shown, extracting only 5-10 minutes was enough to produce similar performance as on the full dataset. (Bottom) The scalability of YASS is shown. As shown, the coreset and clustering algorithms are roughly linear in computational costs with increasing time.

for a single channel or for many channels is simply to use template matching [4]. However, template matching requires having templates that are specific to the recording of interest in advance and does not allow much variability in spike shape. Even in a repeated experiment setting, small changes in the environment, such as shift in electrodes, would change the shape of templates, rendering previously obtained templates unusable. This viewpoint is taken in several approaches, such as [28, 12, 36]. Despite the appeal of these approaches, they are often computationally expensive and difficult to combine with state-of-the-art clustering approaches.

With the increasing popularity of dense MEAs, more complex rules have been proposed to utilize information from all channels simultaneously, such as SpikeDetekt [24]. We note that our methodology is structured in a modular way, such that our pipeline can easily adopt any of these existing methods. However, we also advocate for the development of *data-driven* approaches. In many cases the same device types have been reused for many experiments, and there exists a large collection of example data where false positives and true positives have been thoroughly assessed and curated. In these cases, we will propose a novel approach based on recent advances in deep learning to learn efficient, real-time detection algorithms. This is in contrast to many existing approaches where features are hand-tuned (e.g. threshold, NEO, SpikeDetekt).

There have been some previous data-driven efforts to train a detection algorithm. For example, [25] used hand-curated results from previous sorts to train a neural network, but this was used to classify waveforms rather than as a pure detection method. Furthermore, [45] trained a support vector machine to detect spikes in a simulated recording that provided improvements over a threshold method. Our approach is down a similar line, where we use both previous hand-curated results and synthetic data to train a neural network that dramatically improves the detection quality, as demonstrated in Figure S5.

While this approach is dependent on existing training data and may not be practical everywhere, we emphasize that our pipeline gives state-of-the-art or near state-of-the-art results conditioned on the spike detection method. When curated training data exists (which is true for many research labs),

though, this approach will learn features necessary for detection from the data, and we demonstrate that it can *significantly* improve performance in real data problems. It dramatically reduces the amount of false positives for the same level of true spikes. More importantly, by detecting only well isolated spikes and aligning them properly, it improves the quality of the feature extraction and the signal-to-noise ratio.

## C.2   Neural Network Training Data

The training data for the neural network is constructed from hybrid synthetic and real data based on previous sorts. Training labels used for the training data generation are either defined from a full deconvolution pipeline or a hand-curated effort to validate results. The form of the data is assumed as follows: we have $N_{train}$ time series on a single electrode, where $\mathbf{x}_n \in \mathbb{R}^{R_n}$ reflects the noisy voltage signal and $\mathbf{y} \in \{0, 1\}^{R_n}$ is a binary time series where the value "1" denotes the presence of a well-isolated action potential (i.e. no collisions). We assume that $R_n >> R$, where $R$ is the length of an action potential. The neural network training uses a binary scheme, where any background signals are considered to be false, and the clean spikes are considered to be true.

It is likely that previous sorts do not reflect a perfect ground truth. They may contain false negatives and poorly aligned spikes, which could lead to the creation of faulty training set. As an alternative, synthetic spikes can be constructed from the previous sorts [49, 43]. Therefore, the data used to train the neural network is constructed by a simple method to augment the training set which requires that the mean of clusters must resemble the shape of spikes based upon the procedure in [37]. First, a "noise" training set can be obtained from the noise floor, which is determined by using a low amplitude threshold so that it excludes most of spikes. As a low threshold is used, it is rescaled to match the real noise level presented in the recording. An augmented spike is constructed by superimposing real noise onto randomly-scaled templates. The augmented spikes are randomly placed with the limitation that all spikes must be isloated. This encourages the network to not learn to detect collided spikes, acting as a weak form of triaging. This way, spikes are well aligned and also vary in amplitude.

A potential concern is that the proposed augmentation method may not work well if noise is highly correlated with the spikes, as is believed to be the case in cortical data. Certainly, this issue also exists in the retinal dataset we evaluate on here to some extent. Therefore, based on our experience with retinal data, our hope is that the method will be effective more generally; however, for now this is an empirical question.

## C.3   Neural Network Structure

The architecture of the detection network is a fully convolutional neural network with two hidden layers (see [17] for a background on neural networks). There are $K_1$, $K_2$, and $K_3$ filters of length $R_1$, $R_2$, and $R_3$, respectively, in each of the convolutional filter banks. In our experiment, $R_1 = R$, which corresponds to 2 milliseconds ($K_1 = 60$ for 30kHz recording) and $R_2, R_3 = 5$. $K_1$, $K_2$, and $K_3$ are set to 10, 5, and 1 respectively. The input at the first layer is the electrophysiological time series on a single channel, and the output is the binary labels. Rectified linear unit nonlinearities (ReLU) are used at each hidden layer except for the last layer, which uses a sigmoid nonlinearity $\sigma(x) = \exp(x)/(1 + \exp(x))$ to map the probability to $[0, 1]$. The model was regularized by adding an $l_2$ penalty on the filter weights. The network is learned by minimizing the cross-entropy loss using the Adam update rule [26].

## C.4   Detection using the Neural Network

After the neural network has been learned, it is applied in a channel-wise manner to transform the recorded voltages $\mathbf{V} \in \mathbb{R}^{T \times C}$ into a matrix of probabilities $\mathbf{P} \in [0, 1]^{T \times C}$. An event (isolated spike or collision) is declared if the maximum value of $\mathbf{P}$ is above the detection threshold $\tau$ in a local temporal area. In our experiments, $\tau$ is set to 0.5.

Each returned waveform includes $R' = \frac{R-1}{2}$ samples before and after the spike time (corresponding to $1\ ms$). For each $n$ of the $N$ detections, let $t_n$ and $c_n$ be temporal location and spatial location of the waveform. Each spike is then defined as $\mathbf{X}_n \in \mathbb{R}^{R \times C}$, where the $c^{th}$ channel is defined $\mathbf{x}_{nc} = (V_{t_n - R', c}, \ldots, V_{t_n + R', c})$. Spatial overlaps are handled in a subsequent step, discussed in Section 2.3.

Figure S4: **Illustration of the Neural Network Detection** (Left) The 10 learned filters in the first convolutional layer of the 2-class single-channel neural network. (Right) The neural network transforms a neural recording (Top) into probabilities of spikes (Bottom). Locations of isolated spikes clearly have high probabilities.

Figure S5: **Receiver Operating Characteristic (ROC) curve for Neural Network Detection and SpikeDetekt.** The left panel shows the ROC curve for a cluster with SNR 1.4 and the right panel shows for a cluster with SNR 2.3. Note that instead of the false positive rate, the number of false positive spikes are used for comparison.

## C.5 Empirical Performance and Scalability

The proposed detection algorithm is empirically tested and compared to `SpikeDetekt` [24] on simulated data with high noise level using `ViSAPy`. Since spikes with high enough energy are captured well, cluster-specific ROC curves are plotted. As shown in Figure S5, using the neural network for detection dramatically reduces the number of false positives while keeping the number of true positives reasonably high.

The SNR is defined as $||Template||_2/\sigma\sqrt{|Template|}$, where $Template$ is the mean of all waveforms in the cluster at its highest energy channel and $\sigma$ is the noise standard deviation. $|Template|$ is the number of bins in the template. The true positive rate is defined as the percentage of events that are detected with spatial and temporal alignment to the ground truth. All other events are defined as false spikes.

When the trained neural network is applied, the algorithm scales linearly with the time duration of the sample and the number of channels. As an example of the computational needs of applying this step, on a 5 minute recording of 49 channels, the proposed detection algorithm requires 6 seconds on a single NVIDIA Titan X GPU. Training the network is more time consuming, but not a computational bottleneck. For a 512 channel recording used in our experiments (described in Section 3), the training process required 2 minutes on the same GPU. Compared to the total spike sorting time, as discussed in Section 3.3, training the network is a reasonable cost that quickly becomes trivial as the algorithm is applied to more and more datasets.

# D  Additional Details on Feature Extraction and Masking

The data dimensionality of each spike increases linearly with the number of channels, which naively poses huge computational challenges and renders real-time analysis of large MEAs infeasible. Therefore, it is necessary to utilize spatial *locality*: a given neuron will appear strongly on only a subset of neighboring electrodes. This spatial locality is constructed by restricting detected waveforms to nearby channels via a sparse masking vector, $\{\mathbf{m}_n\} = [\mathbf{m}_{n1}, \ldots, \mathbf{m_{nC}}] \in [0, 1]^C$, as in [24]. This masking vector should be 1 on channels where the neural waveform is reasonably strong (i.e. only the local area), and 0 otherwise. Once the sparse representation is used, the waveform only needs to be considered on channels where the mask is 1, dramatically reducing the effective dimensionality. The effective dimensionality is then dependent on the spatial density of the array and not its total size, so increasing the size of the array does not alter the effective dimensionality. This is a crucial consideration for computational efficiency in many steps of the pipeline, where there is a linear (or worse) scaling with the effective dimensionality.

The construction of the sparse masking vector follows from [24]. First, note that in the feature space a channel that does not have waveform signal present (i.e. a "noise" channel) is expected to follow the normal distribution $\mathcal{N}(0, \mathbf{I}_P)$. This distribution follows from the assumption of whitened background noise. Therefore, the task is to decide whether the signal on each channel is simply a noise event or not. Using a "strong" and "weak" threshold, $\theta_s$ and $\theta_w$, respectively, each mask entry for waveform $n$ in each channel $c$ is determined based on the norm of the power in the feature space, given by

$$
\mathbf{m}_{nc} = \begin{cases} 1 & \text{if } ||\mathbf{y}_{n,c}||_2 > \theta_s \\ 0 & \text{if } ||\mathbf{y}_{n,c}||_2 < \theta_w \\ \frac{||\mathbf{y}_{n,c}|| - \theta_w}{\theta_s - \theta_w} & \text{otherwise} \end{cases} .
$$

Note that there is additional spatial connectivity considered in [24], which is ignored here. In our empirical results, there was no additional benefit to considering spatial connectivity and it added significant extra computational time.

To facilitate efficient inference, the detected waveforms are represented by a "virtual" data distribution, where $\{\tilde{\mathbf{X}}_n\} = [\tilde{\mathbf{x}}_{n1}, \ldots, \tilde{\mathbf{x}}_{\mathbf{nC}}]$, which is given by

$$
\tilde{\mathbf{x}}_{nc} = \begin{cases} \mathbf{x}_{nc} & \text{with probability } m_{nc} \\ \mathcal{N}(0, \mathbf{I}_R) & \text{with probability } 1 - m_{nc} \end{cases} .
$$

Thus, virtual data are given by a mixture of either the original value or a draw from the noise distribution. $R$ is the temporal length of each waveform. Remarkably, the virtual data distribution allows trivial updates during the clustering stage whenever $m_{nc}$ is 0 [24]. This is crucial to make the computational costs in the DP-GMM scale linearly with the number of electrodes and the length of the recording, as discussed in Section 2.6. The same approach is used to construct a virtual data distribution in the feature space, which would be used in the clustering algorithm.

# E  Additional details on the Coreset Construction

There is significant prior work on how to develop a set of representative points that make up a coreset [19, 13]. These prior works developed rigorous approaches with statistical guarantees on representation fidelity. After the development of the coreset, the clustering algorithm is then run on the summary data in time that scales with the size of the coreset, $G$ rather than the raw data size. However, in practice, we found the resulting $G$ to not be large enough to provide reasonable guarantees. Furthermore, following existing strategies with a smaller number of representative points empirically failed to provide coverage over all clusters, as shown in Figure S6.

Thus we provide a simple alternative method to construct the coreset that empirically worked well to capture all visible clusters. The procedure begins by first running K-means++ [1] based on the Euclidean distance with a predefined number of clusters, $K_{++}$. The running time of this step is $\mathcal{O}(dK_{++}N)$ if we assume constant iterations, as is common [1, 3]. $d$ here represents the complete dimensionality of each point, which is $CP$ without considering the effect of the masking vector. To finish the construction of the coreset, if any center in k-means has a associated point that is unacceptably far away (determined by a threshold $D_{\max}$), each cluster is recursively partitioned by

Figure S6: **Illustration of coreset construction.** Simulated data from Gaussian Mixture models. Four different clusters are clearly visible. (Left) Coreset from [2]. The red points are the coreset and their size represents the weight. It is clear that the cluster shapes will not be well represented by the existing coreset method. (Right) Coreset from the proposed method. The sizes of red diamonds represent the weights and the circles are two times standard deviations of each group. As all points contribute to the coreset, the shape of cluster will be well preserved.

---

**Algorithm S1** Constructing the Coreset of Representative Points

---

[representativeWaveforms, sufficientStatistics] $\leftarrow$ `coresetConstruction`(cleanWaveforms)
Input: cleanWaveforms are given by $\mathcal{X} = \{\mathbf{x}_n\}_{n=1,\ldots,N} \in \mathbb{R}^d$
Algorithmic Settings: Distance threshold $D_{\max}$, number of splits $K$, and the maximum
    iteration number $I_{max}$, distance function $D(\cdot, \cdot)$
Output: Representative waveforms and their sufficient statistics
Apply `coreset` (below) to partition the data
Return: centroids (representativeWaveforms) and sufficient statistics of each entry in the partition

Support function: $\{\mathcal{X}_1, \ldots\} = $ `coreset`$(\mathcal{X}, D_{\max}, K)$
% Run initial partition
$\{\mathcal{X}_1, \ldots\} = $ `Kmeans++`$(\mathcal{X}, K, I_{max})$
% Recursively split partitions that are too diffuse
**for** $k = 1, \ldots, K$ **do**
    **if** $\max(D(\mathcal{X}_k, \text{mean}(\mathcal{X}_k))) > D_{\max}$ **then**
        $\{\mathcal{X}_{k1}, \ldots\} = $ `coreset`$(\mathcal{X}_k, D_{\max}, K)$
    **end if**
**end for**
Gather all partitions and reconstruct into $\{\mathcal{X}_1, \ldots\}$
**for** $k = 1, \ldots$ **do**
    $\mathcal{X}_k = \{x_n^k\}_{n=1,\ldots,N_k} \in \mathbb{R}^d$
    sufficientStatistics $= \left( \sum_{n=1}^{N_k} x_n^k, \sum_{n=1}^{N_k} (x_n^k)^T(x_n^k) \right)$
**end for**

---

reapplying K-means++. At the end, only sufficient statistics, mean and covariance of each partition, need to be passed on to DP-GMM described in the next section. The details of this recursive strategy are shown in Algorithm S1.

Unfortunately, this approach is infeasible to run on all channels simultaneously. To address this problem, data are partitioned based on their primary channels (the channel with the highest energy), and only the waveform data on the primary channel and its neighbors are used to construct the coreset. Algorithm S1 is then applied on each set (primary channel) in the partition. These approach reduces the complexity of the primary K-means++ call to $\mathcal{O}(PC_{eff}K_{++}N_c)$, where $C_{eff}$ is the number of neighboring channels and $N_c$ is the number of waveforms in the $c$th partition. Note that $K_{++}$ is set to a smaller value when applied to a single channel, providing a large source of computational savings.

# F  Additional details on the Dirichlet process Gaussian mixture model (DP-GMM)

One of the biggest issues with using GMMs for clustering is that the appropriate number of mixture components $k$ (the number of visible distinct neural signals) is unknown *a priori*. A common approach is to fit a GMM with varying values of $k$ and then perform model selection through the Akaike Information Criterion or the Bayesian Information Criterion [24, 44].

A contrasting approach is the Dirichlet Process Gaussian Mixture Model (DP-GMM), which defines a prior over $k$. There is a rich literature on inferring $k$ using either Markov chain Monte Carlo methods or variational inference; see [48, 9] for previous spike sorting applications. Here, we will first set up the DP-GMM and describe a split-merge Variational inference method to learn the model based on [21]. We will then describe how to alter these algorithms to work with our coreset to facilitate fast and scalable inference. All GMM approaches require inference over a non-convex log-likelihood, where finding optimal parameters is non-trivial. The split-merge approach empirically improves efficiency and finds improved local solutions through efficient search of the parameter space.

When the GMM formulation is used to analyze multiple channels simultaneously, certain modifications need to be made for both statistical and computational considerations. Following [24], we fit a mixture model on the virtual masked data $\{\tilde{\mathbf{Y}}_n\} = [\tilde{\mathbf{y}}_{n1}, \ldots, \tilde{\mathbf{y}}_{n\mathbf{C}}]$, instead of the actual data $\{\mathbf{Y}_n\}$. This allows the mixture model to use the localized nature of the data, which can dramatically reduce computations. Second, following [9, 24], we define the covariance structure in the GMM between channels to be 0. This step reduces the number of parameters to estimate in the covariance matrix to $\mathcal{O}(C)$ from $\mathcal{O}(C^2)$.

This full model can be represented succinctly using a stick-breaking formulation of mixture weights [22] and a Normal-Wishart prior for each cluster and electrode. The assignment variable $z_n$ denotes which cluster the $n$th waveform is assigned to. Letting $k$ define the cluster index, the full generative process of this model is

$$\tilde{\mathbf{y}}_{nc} \mid \{\mu_{\mathbf{kc}}, \mathbf{\Lambda}_{\mathbf{kc}}\}_{k=1,\ldots}, z_n \sim N(\mu_{z_n c}, \mathbf{\Lambda}_{z_n c}^{-1}), \quad z_n \sim \text{Discrete}(\mathbf{w}),$$

$$\{\mu_{kc}, \mathbf{\Lambda}_k\} \sim \mathcal{NW}(\mathbf{m}_0, \lambda_0, \mathbf{W}_0, v_0), \quad w_k = b_k \prod_{\ell=1}^{k-1}(1 - b_\ell), \quad b_k \sim \text{Beta}(1, \alpha_0).$$

We note that if the mean and precision are known for a cluster, then inference for the optimal placement of a waveform exactly follows traditional GMM approaches. Second, we note that the stick-breaking formulation on $\mathbf{w}$ is such that the sum of the probabilities goes to 1 as $k \to \infty$. While this has nice theoretical properties, this is a practically difficult representation, typically requiring the use of a Chinese Restaurant Process formulation [34] or adaptive methods such as Retrospective Sampling [10]. In practice, it is common to simply truncate the maximum value $K$ at a high value.

We alternatively perform inference in this model via by adapting the Variational Bayesian (VB) split-merge approach of [21], which dynamically chooses this truncation level $K$, to utilize our coreset representation. The key idea of the split-merge approach is based on two moves. First, the merge, or cluster death, will combine two clusters if there is not sufficient statistical evidence to support distinct clusters. Vice versa, the split, or cluster birth, will take a cluster that represents an inhomogeneous waveform population and split it into multiple clusters (or neurons).

The first step in the mathematical formulation of this approach is to define approximate posterior forms. Letting $\Theta = \{\mu_{kc}, \mathbf{\Lambda}_{kc}\}_{k=1,\ldots,K,c=1,\ldots,C}$, this is given by

$$q(z, \Theta, \mathbf{b}) = \tag{1}$$
$$\left[\prod_{n=1}^{N} q(z_n | \hat{r}_{n1}, \ldots, \hat{r}_{nK})\right] \left[\prod_{k=1}^{K} q(b_k | \hat{\alpha}_{k1}, \hat{\alpha}_{k0}) \prod_{c=1}^{C} q(\mu_{kc}, \mathbf{\Lambda}_{kc} | \hat{\mathbf{m}}_{kc}, \hat{\lambda}_{kc}, \hat{\mathbf{\Lambda}}_{kc})\right],$$

where $q(z_n = k) = \hat{r}_{nk}$, $q(b_k | \hat{\alpha}_{k1}, \hat{\alpha}_{k0}) = \text{Beta}(\hat{\alpha}_{k1}, \hat{\alpha}_{k0})$, $q(\mu_{kc} | \mathbf{\Lambda}_{kc}) = N(\hat{\mathbf{m}}_{kc}, (\hat{\lambda}_{\mathbf{kc}} \mathbf{\Lambda}_{kc})^{-1})$, and $q(\mathbf{\Lambda}_{kc}) = \delta(\hat{\mathbf{\Lambda}}_{kc})$, a delta function at a particular point. The use of a delta function on the precision is non-standard and breaks the mean-field formulation, but allows us to provide an option to enforce a minimum variance. Enforcing that the minimum cluster variance does not go below the known background variance can improve robustness is certain situations; however, it comes with an increase in computational costs, so the algorithm default is to not use a minimum variance. Precedence of this approximation in VB inference can be found in [21]. Because the DP has infinite mixture components, we implicitly assume that $q(z_n = k) = 0 \ \forall k > K$. The variational parameters

---

**Algorithm S2** Overview of the DPMM procedure on a Coreset.

---

Input: $G$ sufficient statistics for each group, $t_g$, $g = 1, \ldots, G$

Initialize: number of clusters, $K^{(0)}$, global parameters, $\theta_k$, and local parameters for each group and cluster, $\hat{r}_{g,k}$, suff. stat. for each group and cluster, $\mathbf{S}_{g,k}^{(0)} = \sum_{g=1}^{G} \hat{r}_{g,k} t_g$

$\bar{\mathbf{S}}_k = \sum_{g=1}^{G} \mathbf{S}_{g,k}^{(0)}$

**for** i=1,... **do**

    Randomly pick a cluster $k$, split it into $K' + 1$ clusters, yielding $K^{(i-1)} + K'$ clusters

    **for** g=1,..., G **do**

        Update $\hat{r}_{g,k}$ and $\mathbf{S}_{g,k}^{(i)}$

    **end for**

    Update $\theta_k$, for $k = 1, \cdots, K^{(i-1)} + K'$ using $\mathbf{S}_{g,k}^{(i)}$

    Calculate $\tilde{\mathcal{L}}(q)^{(i)}$

    Merge clusters if resulting $\tilde{\mathcal{L}}(q)$ is lower than $\tilde{\mathcal{L}}(q)^{(i)}$. Let $K^{(i)} \leq K^{(i-1)} + K'$ be the resulting number of clusters

**end for**

---

are learned to minimize the KL-divergence between the true posterior and the variational distribution, which maximizes the Evidence Lower Bound Objective (ELBO), given by

$$\tilde{\mathcal{L}}(q) = \mathbb{E}_{\tilde{\mathbf{Y}}|\mathbf{Y}} \left[ \mathbb{E}_q \left[ \log p(\tilde{\mathbf{Y}}, z, \Theta, b) - \log q(z, \Theta, b) \right] \right]. \tag{2}$$

Compared to the typical ELBO used in [21], we have an expectation over $\tilde{\mathbf{Y}}|\mathbf{Y}$ given by the masking approach. While this initially seems like it increases the complexity of the variational updates, the expectation over the mask will simply lead to a linear multiplicative factor when calculating updates. Hence, since the mask is typically 0, this allows great speedups by allowing sparse updates and reduces overfitting.

If the minimum variance constraint is set, then it is necessary to solve $\arg\max_{\tilde{\Lambda}_{kc} \preceq \sigma_{\min}^2 \mathbf{I}} \tilde{\mathcal{L}}(q)$ given the other parameters. Fortunately, a simple procedure can give the optimal solution. Succinctly, the MAP value from the standard variational update on the Wishart distribution is projected to the feasible set on its singular values. The projection is performed by taking the SVD, setting the projected singular values to the minimum of itself and $\sigma_{\min}^2$, and reconstructing. It is straightforward to prove that this update is optimal on the feasible set. A detailed mathematical description of this update and how the standard updates from [21] change is given in the following section. This step gives a modest reduction in overfitting and improves stability at the expense of additional computation.

As discussed in previous sections, running on all data points can lead to slow and redundant computations, so we want to modify the existing VB structure to utilize computations only on the coreset. In mixture models of exponential families, the mean-field parameters for the approximate posterior are determined completely by additive sufficient statistics. Therefore, when working with a coreset, each representative point can store the sufficient statistics of its members that can easily be used when updating variational parameters. Importantly, despite using only a computationally-friendly small set of representative points, these representative points allow the sufficient statistics from each member point to be included in posterior estimates. Specifically, the $G$ representative points, $\{\mathbf{Y}_n^g\}$, their masks, $\{\mathbf{m}_n^g\}$, and their sufficient statistics, $t_g = t(\{\mathbf{Y}_n^g\}, \{\mathbf{m}_n^g\})$ for the $G$ representative points $n = 1, \ldots, N_g, g = 1, \ldots, G$ can be passed to the clustering algorithm to be used in the updates.

Once the ELBO is set up and modified to address masked data and representative points, there is a principled way to choose whether splitting or merging clusters is appropriate. We use the approach of [21], where following an update based off of the coreset, the algorithm proposes splits and merges to search for an optimal point. We give a high-level description of this in Algorithm S2, and further mathematical details in Section F.1.

The default hyperparameter settings used in this algorithm are shown in Table S1, of which the most important are the hyperparameter for the stick-breaking parameter and the hyperparameters for the Normal-Wishart prior. The stick-breaking parameter was set to 1; however, similar performances were obtained from $10^{-1}$ to $10^1$. The hyperparameters for the Normal-Wishart were set such that the

expected covariance of the cluster matches the background noise signal (e.g. $\mathbf{I}$) with a non-informative mean.

## F.1   Additional Mathematical Details

The variational parameters are chosen to (locally) minimize the KL-divergence between the posterior distribution and variational distribution, which is equivalent to maximizing the evidence lower bound (ELBO). However, as the virtual data, $\{\tilde{\mathbf{Y}}\}$, is not directly observed, the expected value of $\mathcal{L}(q)$ given $\{\mathbf{Y}_n\}$ is maximized. Accordingly, the following objective function is optimized:

$$\tilde{\mathcal{L}}(q) = \mathbb{E}_{\tilde{\mathbf{Y}}|\mathbf{Y}}\big[\mathbb{E}_q\big[\log p(\tilde{\mathbf{Y}}, z, \mu, \mathbf{\Lambda}, b) - \log q(z, \mu, \mathbf{\Lambda}, b)\big]\big]$$

The local parameters given global parameters are updated as:

$$\rho_{nk} = \exp(\mathbb{E}_{\tilde{\mathbf{Y}}|\mathbf{Y}}[\mathbb{E}_q[\log p(\tilde{\mathbf{Y}}_n|\mu_k, \mathbf{\Lambda}_k) + \log w_k]]), \quad \hat{r}_{nk} = \frac{\rho_{nk}}{\sum_{j=1}^K \rho_{nk}}$$

Given sufficient statistics, $\hat{N}_k = \sum_{n=1}^N \hat{r}_{nk}$, $\tilde{\mathbf{s}}_{k1} = \sum_{n=1}^N \hat{r}_{nk}\mathbb{E}_{\tilde{\mathbf{Y}}|\mathbf{Y}}[\tilde{\mathbf{Y}}_n]$, $\tilde{\mathbf{s}}_{k2} = \sum_{n=1}^N \hat{r}_{nk}\mathbb{E}_{\tilde{\mathbf{Y}}|\mathbf{Y}}[\tilde{\mathbf{Y}}_n\tilde{\mathbf{Y}}_n^T]$, the global parameters are updated as following:

$$\hat{\alpha}_{k1} = 1 + \hat{N}_k, \quad \hat{\alpha}_{k0} = \alpha_0 + \sum_{j=k+1}^K \hat{N}_j, \quad \hat{\lambda}_k = \lambda_0 + \hat{N}_k, \quad \hat{\mathbf{m}}_k = \frac{1}{\hat{\lambda}_k}(\lambda_0\mathbf{m}_0 + \tilde{\mathbf{s}}_{k1}), \quad \hat{v}_k = \hat{N}_k + v_0$$

Updating $\hat{\mathbf{\Lambda}}_k$ is a two-step process. $\hat{\mathbf{\Lambda}}_k$ is first updated to maximize expected ELBO.

$$\hat{\mathbf{\Lambda}}_k = (\hat{N}_k - P - 1)\Big(\mathbf{W}_0^{-1} + \tilde{\mathbf{s}}_{k2} - \frac{1}{\hat{N}_k}\tilde{\mathbf{s}}_{k1}\tilde{\mathbf{s}}_{k1}^T + \frac{\lambda_0\hat{N}_k}{\lambda_0 + \hat{N}_k}\Big(\frac{\tilde{\mathbf{s}}_{k1}}{\hat{N}_k} - \mathbf{m}_0\Big)\Big(\frac{\tilde{\mathbf{s}}_{k1}}{\hat{N}_k} - \mathbf{m}_0\Big)^T\Big)^{-1}$$

Then, let $\hat{\mathbf{W}}_k = \mathbf{A_k}\mathbf{\Sigma_k}\mathbf{B_k^T}$, which is a singular value decomposition. The second update is

$$(\mathbf{\Sigma_k})_{ii} = \begin{cases} 1 & \text{if } (\mathbf{\Sigma_k})_{ii} > 1 \\ (\mathbf{\Sigma_k})_{ii} & \text{if } (\mathbf{\Sigma_k})_{ii} \leq 1 \end{cases}$$

It ensures that the variance of each component is bigger than 1 as the variance is the sum of cluster variance and noise variance, which is one.

To further simplify the process, the independence of data across the channels can be assumed. Then, the covariance matrix of cluster has a block-diagonal shape and the cluster shape in each channel can be estimated separately using the above update. This is reasonable assumption as spatially whitened data is used.

The split and merge steps are conceptually the same as in [21], but adjusted to use the coreset.

## G   Addditional details on the Divide and Conquer approach

Because of the method used to divide the data in the divide-and-conquer step, it is possible that the same neuron may have templates and clusters under different spatial subsets. Furthermore, the GMM approach may overcluster due to model mismatch (e.g. non-Gaussianity of the clusters). Therefore, it is necessary to merge the templates prior to the deconvolution step.

The templates are constructed as follows. After the clustering process, each spike $\mathbf{X}_n$ has been associated with one of $K$ clusters, which is denoted by an assignment variable, $z_n \in \{1, \dots, K\}$. Then, let $\mathbf{W}_k \in \mathbb{R}^{R \times C}$ be defined as the mean of each cluster defined as $mean(\{\mathbf{X}_n|z_n = k\})$, which is taken with respect to the original data rather than the virtual data distribution. In addition, a binary mask is defined for each cluster by $\mathbf{m}_k^w \in \{0, 1\}^C$, where

$$m_{kc}^w = \begin{cases} 1, & \text{mode}(\{m_{nc}|z_n = k\}) > \theta_w^{mask} \\ 0, & \text{otherwise} \end{cases}.$$

**Algorithm S3** Template Merging

---

templates $\leftarrow$ mergeTemplates($\{$clusterAssignments$_i\}_{i=1,\ldots}, \{$representativeWaveforms$_i\}_{i=1,\ldots}$)
Algorithmic Settings: Maximum shift allowance $s_{max} \in \mathbb{N}$, thresholds, $\theta_1, \theta_2 \in (0,1)$, number of clusters, $K$, temporal length of waveform, $R$, number of channels, $C$.
Input: clusterAssignments are given by $\{z_n\}_{n=1,\ldots,N} \in \{1,\ldots,K\}$ and representativeWaveforms are given by $\{\mathbf{X}_n^s\}_{n=1,\ldots,N,s=-s_{max},\ldots,s_{max}} \in \mathbb{R}^{C \times R}$, waveforms shifted by $s$ from its center.
Output: templates

% Get templates with all shifts
**for** $k = 1,\ldots,K$ **do**
  $\mathbf{W}_k^s = mean(\{\mathbf{X}_n^s | z_n = k\})$
  $w_k = |\{\mathbf{X}_n^s | z_n = k\}|$
**end for**

Initialize: Undirected Graph $G$ with $K$ nodes and 0 Edges
**for** $k_1, k_2 = 1,\ldots,K$ **do**
  % shift that maximizes the cosine of two templates
  $s' = \arg\max_s cosineDist(\mathbf{W}_{k_1}^0, \mathbf{W}_{k_2}^s)$
  **if** $cosineDist(\mathbf{W}_{k_1}^0, \mathbf{W}_{k_2}^{s'}) < \theta_1$ and $\theta_2 < \frac{||\mathbf{W}_{ck_1}^0||_2}{||\mathbf{W}_{ck_2}^{s'}||_2} < \frac{1}{\theta_2}, c = 1,\ldots,C$ **then**
    Add edge on $(k_1, k_2)$
  **end if**
**end for**
Using strongly connected components on $G$ [42], templates are grouped into $K'$ groups.
For $k = 1,\ldots,K'$, new template $\tilde{\mathbf{W}}_k$, is the mean of properly shifted templates in each group weighted by $w_k$.

---

The mode on this continuous distribution is estimated by the peak of a kernel density estimate. $\theta_w^{mask}$ is set to 0.5. Given template masks, templates can be localized by considering channels with only non-zero mask entries. Template merging is performed based on two criteria, the angle and the amplitude of templates. When two templates are in fact the same neuron, the template waveforms should have a similar shape and their amplitudes should approximately match. To determine whether the waveforms have a similar shape, typically the cosine distance is used; however, the cosine distance is greatly affected when templates are not temporally aligned. Therefore, the "angle" is calculated as the cosine distance on the best alignment between two templates. After the similarity check, an undirected graph can be created by considering each template as a node and constructed edges based off of similarity. Once the graph is obtained, strongly connected components are computed using Tarjan's algorithm [42] to group templates. Then, new templates are created by taking the mean of each group. The pseudocode to perform this merging is shown in Algorithm S3.

# H   Constructing a proxy to ground truth for a real dataset

It is notoriously difficult to obtain ground truth data for large-scale extracellular recordings of neural activity with MEAs. However, in rare cases, sufficiently strong anatomical and functional priors are available for the tissue under study and make it possible to hand-curate the outcome of a sorting pipeline, resulting in an acceptable partial proxy to ground truth on these data.

The data we used to construct such a proxy consisted of 30 minutes of extracellular recordings of the peripheral primate retina, obtained ex-vivo using a high-density 512-channel recording array [30]. During the half-hour recording, the retina was stimulated with spatio-temporal white noise in which a lattice of square pixels were updated randomly and independently of one another over time. The intensity of each display primary at each pixel location was chosen from a binary distribution at each refresh, yielding a stimulus with chromatic variation.

The reference set of spike times was manually assembled as follows. Events whose amplitude exceeded 4 times the RMS noise on each electrode were detected, aligned to the time of peak deviation from baseline using cubic spline interpolation for sub-sample alignment, and noise-whitened.

Principal component analysis was performed on the collection of events detected on each electrode and the surrounding 6 electrodes, and the collection of events was clustered using a variation of the Ng-Jordan-Weiss spectral clustering algorithm [35, 52]. The hyper-parameters of the clustering algorithm were swept, resulting in a large collection of candidate neurons. Candidate neurons whose spike train exhibited significant violations of the refractory period were rejected.

Reference neurons were then identified on the basis of their light response properties. Anatomical and functional priors guarantee that classes of retinal ganglion cells in the primate retina tile the visual field uniformly, forming well-coordinated mosaics [11]. After calculating the spike-triggered average response of each neuron, cells were therefore separated in unique functional types, corresponding to the ON and OFF midget cells, ON and OFF parasol cells, ON and OFF upsilon cells, small bistratified cells, and polyaxonal amacrine cells. For each cell detected more than once in a mosaic, only the cell with the largest spike count was kept. All cells with physiological properties incompatible with the anatomical and functional priors of the primate retina were discarded from the analysis, resulting in a final collection of unique reference neurons of known cell types (n = 355), as well as a few neurons whose physiological properties were consistent with yet unreported cell types (n = 31).

Note that, by construction, this gold standard database includes a collection of trusted spike shapes and times, but we do not claim that this analysis captures every single true spikes from this recording; indeed, we expect that the dataset includes a number of unlabeled small or collided spikes.

## I  Relationship to existing pipelines

Prior to the clustering stage, several important preprocessing steps take place. Filtering with wavelets [47] can improve sorting performance; however, we chose to use a bandpass Butterworth filter [29] because it is used in most academic and commercial implementations, such as the Plexon Offline Sorter™.

The most common detection algorithm selects events that cross a threshold, with alignment to the threshold crossing or a peak [29]. Alternatively, [24] used multiple thresholds and temporal and spatial adjacency to improve detection, and aligned to the mean energy of a soft thresholded signal; [38] used wavelets to denoise and detect spikes. All these methods generally lead to many false positives that negatively affect the clustering step. Our pipeline improves on these issues by triaging false positives and addressing them in a later post-processing step (see Section C.1).

The clustering stage of the spike sorting problem has been discussed in several reviews [41, 29, 16, 39]. Spike sorting on dense MEAs has been explored via template matching [15], blind deconvolution [8, 37, 12], and clustering [24, 9]. Our approach is based on the Dirichlet process Gaussian mixture model (DP-GMM), which first introduced to the spike sorting problem by [48]. There have been significant improvements in efficient inference for the DP-GMM, and our work heavily utilizes the memoized approach of [21], but several alternatives exist, including online approaches [46], utilized in [8], and parallel computing [7].

Compared to prior work, we directly address collisions and outliers by excluding them from the clustering step with triaging, which enables significantly improved clustering performance. Furthermore, local minima are a critical problem in many clustering, template matching, and blind deconvolution approaches due to the non-convex nature of these approaches. While our model is also non-convex, our empirical results (not shown) demonstrated that adapting modern variational inference techniques improved both reliability and accuracy. These results match the conclusions of [21]. This is of crucial importance as the field moves towards millions of distinct waveforms, because common hand-sorting and hand-correction steps are intractable on such data.

There are a number of alternative approaches to the clustering step in the literature. Many of these approaches address the Gaussianity assumption on the shape of the clusters. One of the most common alternative approaches to a Gaussian mixture model formulation for clustering is a density based approach. This has been done in the form of superparamagnetic clustering [38], consensus clustering [14], unimodal clustering [31], spectral clustering, finding density peaks [40, 51], and the mean shift algorithm [20]. `MountainSort` [31] uses a non-parametric clustering step that assumes the clusters are unimodal in the sense that they have a single point of maximal density when projected onto any line. These approaches hold significant promise for dealing with non-Gaussian waveform clusters, and many of these clustering methods could be interchanged with our DPMM in our complete

pipeline to alter the clustering step. However, as empirically demonstrated in our experiments, while these approaches can deliver improved performance on non-Gaussian clusters, on our metrics, for the datasets analyzed here, the average performance is below the proposed DPMM.

Recently, the `Kilosort` algorithm [36] was proposed to perform clustering and inference directly on the time series instead of using clustering. Collisions generally cause problems in the PCA and clustering space [12]; the matching pursuit approach of [36] sidesteps the clustering step entirely. In our pipeline, we address the collision problem with our triage-then-pursuit strategy; this lets us use fast clustering primitives to estimate templates, leading to significant scalability gains relative to full matching pursuit approaches. (Clustering approaches are also better able to handle uncertainty in template shape and spike assignments than greedy matching pursuit approaches.) After we have an appropriate clustering and waveform shape estimate, we use some of the methods proposed in [36] and [12] for the final collision-unmixing step. Other methods implicitly incorporate overlapping spike detection in their model training [8]; however, this approach has not yet been demonstrated to scale to large dense MEA datasets.

`Spyking Circus` [51] is another algorithm that aims to scale spike sorting to large MEA recordings by density estimation clustering of dimensionally reduced spikes by PCA, followed by template estimation and matching. Spikes are detected according to threshold crossings on each channel which can affect PCA projections negatively because of collisions and alignment issues. The algorithm is scaled to large recording sizes by sub-sampling spikes (data points), which can under represent units with lower firing rates. This method allows for parallelizing the computation of affinity matrices of data points which has lead to GPU and multi-threading implementations. However a great deal of post processing of results of each channel is needed that in practice renders the method time inefficient compared to `YASS` and `KiloSort`.

`JRClust` [23] follows a similar method as `Spyking Circus`. It does not use deconvolution to infer collisions. The method, however, addresses an issue of non-stationarity caused by noise and probe drift. Due to methodological similarity to `Spyking Circus`, it is expected that their performances are comparable. We hope to provide more detailed comparisons in the future.