[Reviews · NeurIPS 2017]

Reviewer 1



[UPDATE AFTER AUTHOR RESPONSE] The authors' response confirms my rating – it's a valuable paper. I'm optimistic that they will address mine and the other reviewers' concerns in their revision. [ORIGINAL REVIEW] The paper describes YASS: Yet Another Spike Sorter, a well-engineered pipeline for sorting multi-electrode array (MEA) recordings. The approach is sound, combining many state-of-the-art approaches into a complete pipeline. The paper is well written, follows a clear structure and provides convincing evidence that the work indeed advances the state of the art for sorting retinal MEA data. While the paper is already a valuable contribution as is, I think it has much greater potential if the authors address some of the issues detailed below. In brief, my main concerns are (1) the lack of publicly available code, (2) the poor description of the neural network detection method, (3) issues with applying the pipeline to cortical data. Details: (1) Code is not available (or at least the manuscript does not provide a URL). Since spike sorting is at this point mainly an engineering problem (but a non-trivial one), a mere description of the approach is only half as valuable as the actual code implementing the pipeline. Thus, I strongly encourage the authors to go the extra mile and make the code available. (2) Neural network spike detection. This part seems to be the only truly innovative one (all other components of the pipeline have been described/used before). However, it remains unclear to me how the authors generated their training data. Section C.2 describes different ways of generating training data, but it is not clear which one (or which combination) the authors use. (a) Using pre-existing sorts. First, most labs do *not* have existing, properly sorted data available when moving to dense MEAs, because they do not have a pipeline for sorting such array data and – as the authors point out – existing methods do not scale properly. Second, it is not clear to me how existing sorts should help training a more robust neural network for detection. Did the authors inspect every single waveform snippet and labeled it as clean or not? If not, based on what algorithm did they decide which waveform snippets in the training data are clean? Why do they need to train a neural network instead of just using this algorithm? How do they deal with misalignments, which create label noise? If using pre-existing sorts is what the authors did, they need to provide more information on how exactly they did it and why it works. In the current form, their work cannot be reproduced. (b) Generating synthetic training data by superimposing waveform templates on background noise. This could be a reasonable approach. Is it used for data augmentation or not at all and just described as a potential alternative? What is the evidence that this approach is useful? The synthetic data may not be representative of real recordings. (3) Generalization to cortical data. I am quite confident that the pipeline works well for retinal data, but I doubt that it will do so for cortical data (some arguments below). I think this limitation needs to be discussed and acknowledged more explicitly (abstract, intro, conclusions). (a) In cortical recordings, waveform drift is a serious issue that arises in pretty much all non-chronic recordings (and chronically working high-density MEAs are still to be demonstrated). Thus, modeling drift is absolutely crucial for recordings that last longer than a few minutes. (b) Getting good training data for the NN detection is more difficult. Good ground truth (or well validated data such as described in appendix I) is not available and generating synthetic data as described in C.2 is not necessarily realistic, since background noise is caused by spikes as well and neurons often fire in highly correlated manners (thus rendering the approach of overlaying templates on spike-free noise problematic). Minor comments: - Fig. 3 bottom panel: Y axis is strange. Does 1^-x mean 10^-x? Also, it exaggerates tiny accuracy differences between 0.99 and 0.999, where both methods are essentially perfect. - The authors use spatially whitened data (according to section 2.1), but I did not find a description of the spatial whitening procedure in the manuscript or supplement.

Reviewer 2



[UPDATE AFTER AUTHOR RESPONSE] I have upgraded by assessment and confidence about this paper. The upgrade is due to the authors' commitment to including a performance breakdown in the paper, should it be accepted. I would also urge the author's to explicitly mention that their method has supervised stages, whereas some of the comparison methods (i.e., Kilosort) are unsupervised. I agree with the authors' response that if prior information is available, it should be utilized. However, since this is an important methodological detail relevant to the comparisons presented, it must be made explicit to facilitate the community's interpretation of the results. [ORIGINAL REVIEW] The authors develop a multistage spike sorter, cutely coined YASS, which follows the classical pipeline of spike detection, feature extraction, and clustering. The most appealing aspect of this work is that the developed YASS system outperforms relative to several state-of-the-art methods. Despite this laudable performance, this work raises several concerns, as enumerated below. Major Concerns: 1. Yass' impressive performance comes at the cost of requiring hand-labeled data (for the detection step), which is not required for all methods to which Yass is compared. In this sense, the comparisons provided are not as apples-to-apples. Requiring the experimenter to provide such hand-labeled data represents an additional burden of effort which is not required by completely unsupervised methods. 2. Many of the components in the Yass pipeline are derivative of previous work. While there are certainly novel components (possibly enough to merit publication at NIPS), the authors do not provide a performance breakdown to demonstrate that these novel contributions are required to achieve the stated levels of performance. For example, it might be the case that the overall performance is due to the particular selection of previously developed components, rather than due to the inclusion of newly developed ones. Minor concerns 3. If my read was correct, the proposed method discards signals that appear on multiple electrodes, keeping only the waveform signal with the largest amplitude (lines 138-140). This is in contrast to the approach of Kilosort, for example. This seems disadvantageous, and raises another flag that the comparison may not be apples-to-apples against Kilosort. If the experimental data did not contain overlapping signals across channels, Yass' performance would not suffer due to this design choice, and Kilosort's performance would not shine in comparison since one of it's main features would not be in play. If my understanding is not correct here, this text should be clarified. Regardless, the authors should clarify the precise application setting with regard to overlapping signals across channels. 4. The writing style is rather opaque. I would encourage the authors to clarify their presentation by boiling down the critical mathematical and algorithmic details (be they words or equations), and including them in the main paper. At present, almost none of these critical details are available without diving into supplements. This separation of the details presents quite a burden to the reader and the reviewer.

Reviewer 3



This paper addresses the important contemporary problem of spike detection and sorting on multi-electrode arrays (MEA). Efficient algorithms for this problem are critical given the increased number such experiments and the number of simultaneously recorded neurons. The ability to scale into the regimes of 10^4 to 10^6 electrodes is challenging and requires sophisticated analysis methods. The authors have correctly identified key attributes of robustness, scalability, modularity and the use of prior information in the development of powerful algorithms. The paper employs a substantial amount of methods and machinery that the authors are well familiar with and understand the problem space well. Algorithm 1 placed early in the paper is not particularly helpful or readable, and doesn’t really offer anything that the text description doesn’t provide. The overview section is important to approach first but the writing here is not as clear as it might be and moves over a wide range of topics very quickly. It is very difficult to determine the real veracity of the algorithms proposed. In particular the neural net training stage is compelling but difficult to penetrate how it would really work. Having training data, potentially different for each labs use, may be problematic and it is difficult to gauge how well this will really work, although the authors provide some good simulations. In summary, this is potentially an interesting approach although the complexity of the method, its seemingly ad hoc nature and its comparatively not transparent presentation make it less attractive. The approach is interesting although very complicated with technical justification absent from the main paper or relegated to the Supplementary materials, which are substantial.